# Seafood Processing, Preservation, and Analytical Techniques in the Age of Industry 4.0

Abdo Hassoun [1,2,*], Shahida Anusha Siddiqui [3,4], Slim Smaoui [5], İlknur Ucak [6], Rai Naveed Arshad [7], Paula Garcia-Oliveira [8,9], Miguel A. Prieto [8,9], Abderrahmane Aït-Kaddour [10], Rosa Perestrelo [11], José S. Câmara [11,12] and Gioacchino Bono [13]

1    Sustainable AgriFoodtech Innovation & Research (SAFIR), 62000 Arras, France
2    Syrian Academic Expertise (SAE), Gaziantep 27200, Turkey
3    Department of Biotechnology and Sustainability, Technical University of Munich, 94315 Straubing, Germany; shahidasiddiqui777@gmail.com
4    German Institute of Food Technologies (DIL e.V.), 49610 Quakenbrück, Germany
5    Laboratory of Microbial, Enzymatic Biotechnology and Biomolecules (LBMEB), Center of Biotechnology of Sfax, University of Sfax-Tunisia, Road of Sidi Mansour Km 6, P.O. Box 1177, Sfax 3018, Tunisia; slim.smaoui@cbs.rnrt.tn
6    Faculty of Agricultural Sciences and Technologies, Nigde Omer Halisdemir University, Nigde 51000, Turkey; ilknurucak@ohu.edu.tr
7    Institute of High Voltage & High Current, UniversitiTeknologi Malaysia, Skudai 81310, Malaysia; rainaveed@yahoo.co.uk
8    Nutrition and Bromatology Group, Analytical and Food Chemistry Department, Faculty of Food Science and Technology, University of Vigo, Ourense Campus, E-32004 Ourense, Spain; paula.garcia.oliveira@uvigo.es (P.G.-O.); mprieto@uvigo.es (M.A.P.)
9    Centro de Investigação de Montanha (CIMO), Instituto Politécnico de Bragança, Campus de Santa Apolonia, 5300-253 Bragança, Portugal
10   VetAgroSup, INRAE, Université Clermont Auvergne, UMRF, 15000 Aurillac, France; abderrahmane.aitkaddour@vetagro-sup.fr
11   CQM—Centro de Química da Madeira, Campus da Penteada, Universidade da Madeira, 9020-105 Funchal, Portugal; rmp@staff.uma.pt (R.P.); jsc@staff.uma.pt (J.S.C.)
12   Departamento de Química, Faculdade de CiênciasExatas e Engenharia, Campus da Penteada, Universidade da Madeira, 9020-105 Funchal, Portugal
13   Institute for Biological Resources and Marine Biotechnologies, National Research Council (IRBIM-CNR), Via L. Vaccara 61, 91026 Mazara del Vallo, Italy; gioacchino.bono@cnr.it
*    Correspondence: a.hassoun@saf-ir.com

**Abstract:** Fish and other seafood products are essential dietary components that are highly appreciated and consumed worldwide. However, the high perishability of these products has driven the development of a wide range of processing, preservation, and analytical techniques. This development has been accelerated in recent years with the advent of the fourth industrial revolution (Industry 4.0) technologies, digitally transforming almost every industry, including the food and seafood industry. The purpose of this review paper is to provide an updated overview of recent thermal and nonthermal processing and preservation technologies, as well as advanced analytical techniques used in the seafood industry. A special focus will be given to the role of different Industry 4.0 technologies to achieve smart seafood manufacturing, with high automation and digitalization. The literature discussed in this work showed that emerging technologies (e.g., ohmic heating, pulsed electric field, high pressure processing, nanotechnology, advanced mass spectrometry and spectroscopic techniques, and hyperspectral imaging sensors) are key elements in industrial revolutions not only in the seafood industry but also in all food industry sectors. More research is still needed to explore how to harness the Industry 4.0 innovations in order to achieve a green transition toward more profitable and sustainable food production systems.

**Keywords:** fourth industrial revolutions; fish; new preservation methods; emerging technologies; thermal and nonthermal processing

## 1. Introduction

The food industry, as other industries, has evolved overtime and underwent revolutions along the whole food supply chain, including technology and processes involved in food production and consumer demands. Starting from the very beginning of gathering or hunting to using stem cell technology, 3D printing and other advanced techniques to produce any desired food products, technology involved in human nutrition has evolved considerably over time but, the best is yet to come. With increasing customer demands, global health, and environmental crisis, there has been an urgency of developing more sustainable, reliable, and resilient technologies, and these necessities have pushed academic and industrial actors to come up with better innovations [1,2].

The first revolution (late in the 18th century) led the foundation of mechanized systems in the industries fulfilled by steam engines, which were followed by development of rapid processes and mass production during the 2nd revolution in late 19th century, using the oil and electricity. The third revolution is known to have established the foundations of automated systems, using computers and robotics in and after 1960s [3,4]. The fourth revolution has started recently and emerged as a result of the convergence of physical, biological, and digital domains [5]. The fourth industrial revolution (called Industry 4.0 or 4IR) has been successful in increasing the food production quantity as well as quality [6,7]. For example, smart sensors, as an element of 4IR, can be used to sort products according to their physical, chemical, and other properties (this will be more explained later in this paper).

The major technologies used in the 4IR age include artificial intelligence (AI), machine learning (ML), robotics, big data analysis, smart sensors, among many others [8,9]. Microwaves, ultrasound, high pressure based-technologies, biotechnology, nanotechnology, spectroscopy, spectrometry, chromatography, highly sensitive computerized sensors, etc., are examples of technological innovations that are under development and some of them have been already implemented in the food industry. The ongoing pandemic situation caused due to the coronavirus has also contributed in the rapid development and acceleration of the 4IR and adoption of its technologies [10].

Seafood is generally recognized as a vital component of a balanced and nutritious diet due to its low-fat level and high protein content, as well as a variety of micronutrients such as vitamins and minerals. In addition, seafood is the principal dietary source of long-chain polyunsaturated fatty acids (LC-PUFAs or Omega-3), such as eicosapentaenoic acid (EPA) and docosahexaenoic acid (DHA), which give a variety of health advantages, including a lower risk of cardiovascular disease [11,12]. However, seafood is particularly perishable and degrades faster than other foods [13–16]. Traditionally, a range of preservative approaches (such as curing, drying, fermentation, smoking, and conventional storage procedures; refrigeration and freezing) has been commonly applied to fish and other seafood [14,17,18], but they are not without limitations. Some of the challenges associated with the aforementioned traditional techniques include limited preservative usefulness, negative impact of sensory properties, high energy consumption, and hence, harmful effect on the environment. Many advanced technologies, such as ultrasound, irradiation, high-pressure processing, cold plasma, radiofrequency, pulsed electric field, pulsed light technology, microwave processing, and packaging technologies (e.g., modified atmosphere, active, and intelligent packaging) have emerged in recent years with the advent of the 4IR technologies and have the potential to address the above referred challenges and to increase the shelf life of fish and other seafood during processing/and or preservation [19–23].

Safety, authenticity, traceability, nutritional quality, availability, convenience and integrity, and freshness are all important aspects of seafood quality [24–26]. Conventional measurement methods of these quality aspects have been widely reported in the literature to determine sensory, physical, chemical, and microbial quality attributes [23,27]. Traditional measurement methods (such as Kjeldahl method, colorimeter, texture analyzer, plate colony counting, etc.) have been extensively used to determine specific target attributes (such as chemical composition, color, texture, general appearance, biogenic amines, lipid and

protein oxidation products, total volatile basic nitrogen, and microbial growth, among others). However, such analytical approaches are unlikely to be suitable for food Industry 4.0 due to their well-known limitations (e.g., time consuming, destructive, etc.). Therefore, a variety of advanced techniques has been developed in recent years, including among others, various kinds of spectrometry, spectroscopic smart sensors, and hyperspectral imaging techniques [24,26,28].

Current review papers reporting on the use of Industry 4.0 technologies in the seafood industry are limited. Recently, a short overview of particular Industry 4.0 technologies in the food industry has also been given by Chapman and co-authors [29], whereas Jambrak and others reviewed some of the Industry 4.0 platforms, such as AI, big data, and smart sensors [2]. This review paper will give an updated overview of recent use of Industry 4.0 technologies in seafood processing and preservation, as well as analytical operations.

## 2. The Fourth Industrial Revolution

The industrial revolutions can also be divided into four time periods (Table 1) although the dates for the beginning and the end of each industrial revolution are still very much in debate because of the variety and uneven industrial development in different countries; the first revolution started around the end of 18th century and involved use of liquid or steam powered machines to perform the heavy labor duties; the second revolution, involved use of first electrical machines, soon after 30 years of 1st revolution; the third revolution, which started around 1960s used renewable energies and internet; and fourth revolution, started only after 2015 and introduced interdisciplinary strategies that draw on physical, biological, and digital innovations [5]. There has been a huge impact of these four revolutions on the food industry across the globe [5,30–32];

**Table 1.** Industrial revolutions, their characteristics, and their impact on food industries.

| Industrial Revolution | Time Frame | Energy Source | Characteristic Technology | Impact on the Food Industry |
|---|---|---|---|---|
| 1IR | End of 18th century | Coal | Steam engine | First ever mechanized processes, sterilization using steam and liquids, preservation practices. |
| 2IR | End of 19th century | Oil and electricity | Mechanical assembly lines | Scaled up mass production, demands met quality, quantity enhanced. |
| 3IR | 1960s to 2000s | Nuclear energies | Computers, robotics | More scaled up production, computerized and automated processes, high energy consumption—surfacing limits. |
| 4IR | 2015 to Present | Renewable (green) energies | IoT, ML, AI, 3D printing, nanotechnology, biotechnology, smart sensors | Sustainable practices, improved systems, skilled labor, time, space and money saving techniques, real time monitoring. |

Abbreviations: IoT, Internet of Things; ML, Machine Learning; AI, Artificial Intelligence.

First revolution (1IR): The use of machines based on steam or liquid (mostly water) created the groundwork for mechanized production instead of manual or animal power. The 1IR allowed to reduce costs, time as well as raw material wastes, and enhanced sanitization processes. Coal was used as a source of energy for the production of power for steam engines.

Second revolution (2IR): Involved introduction of two new concepts, namely the division of labor and the use of electricity and oil as the new sources of energy. The replacement of steam engines by the electric power allowed scaling up production, enabling mass production. There was also seen a synthesis of chemicals, used in coloring, fertilizers and fabrics and use of food freezing, drying, and pasteurization systems for better production and customer satisfaction.

Third revolution (3IR): During this period renewable (nuclear) energies were used, which revolutionized the production, making processes more sustainable and feasible. The use of electronics, computational systems, and some robotics accelerated this revolution. This led to miniaturized production of materials, which indeed resulted in establishments and beginning of biotechnology, and nanotechnology. The revolution also enhanced the production rates and increased productivity at industrial scale by thousand folds.

Fourth revolution (4IR): As the technology advanced and the global demands increased, non-renewable energy sources started depleting, acknowledging the urgent need for sustainable energy solutions, which led to developments of greener energies, combining the powers of physical, biological, and computational worlds. Emerging technologies, such as genetic engineering, biotechnologies, nanotechnologies, robotics, and many others have been used for the production of high quality and quantity products, with personalized touch for different groups of customers. Stem cell technology to produce meat, test tube meat, probiotics, and vegan and plant-based food products are some of the examples that characterize the current revolution in the food industry [5,33,34]. The convergence between the aforementioned emerging technologies, occurring during the ongoing 4IR, is opening up unprecedented opportunities for the seafood industry. As aquaculture is a growing segment of the seafood sector, European Commission, coined the term aquaculture 4.0, in its Horizon 2020 Innovation Action Call in 2017, describing the reach of Industry 4.0 in the seafood industry.

The use of AI, robotics, and other Industry 4.0 elements in the management, production, processing, packaging and distribution of seafood can be seen as a good example. There has recently been unprecedented development of different computer-based sensors and software, applications and complex robotic systems for regulating, real time monitoring, measuring, and managing various processes in the seafood sector efficiently and effectively. For example, AI was used to develop innovative solutions to make the fishing industry more sustainable in the context of FishFace project (see https://www.natureaustralia.org.au/what-we-do/our-priorities/oceans/ocean-stories/fishface/ (accessed on 27 November 2021)). Image database was created to be used by a software equipped AI to assign fish to a certain species on the basis of photos. Another example is Fishcoin project where blockchain technology is used to trace fish from the point of harvest to the point of consumption (see https://fishcoin.co/ (accessed on 27 November 2021)). Another application of Industry 4.0 technologies in seafood is OxyForcis system that uses an automated optical dissolved oxygen sensor for real time monitoring of oxygen levels in the aquaculture tanks, displaying available data visually through a mobile phone, for example (see https://smalletec.com/oxyforcis-2/ (accessed on 27 November 2021)). Intelligent robotics and smart miniaturized sensors are also among the main features of the Industry 4.0 era that have established a greater automation in several industries [35–37]. For example, Balaban and others described the use of specialized robotics for different tasks (e.g., sorting/handling and processing) in seafood industry [38]. The increasing demand for protein and the recent biotechnological advances emerged with the arrival of Industry 4.0 era have motivated many companies to develop cell-cultured seafood products [39,40]. One example of these companies is BlueNalu (see https://www.bluenalu.com/ (accessed on 27 November 2021)) that produces whole muscle, cell-cultured seafood products directly from living cells isolated from fish tissue.

However, it should be stressed that most of the aforementioned applications have not yet reached the industrial level. More collaboration between laboratory research institutes and industrial actors is indispensable in the future to capture the full potential of Industry 4.0 and to further harness its technologies to solve current challenges.

## 3. Current Trends and Advancements in Seafood Preservation Techniques

Seafood products are susceptible to loss of quality due to different phenomena, such as degradation of muscle proteins, oxidative processes and microbial spoilage, if they are not properly preserved [41,42]. Several traditional techniques were used to preserve these

foods, such as ice, drying, smoking, fermentation, or salting. However, the preservation techniques keep evolving and new strategies have emerged (Table 2).

**Table 2.** Examples of studies applying current trends in seafood preservation techniques.

| Seafood | Preservation Technique | Results | Reference |
|---|---|---|---|
| Tilapia fish (*Oreochromis aureus*) | ICF −5 °C, 0.9% saline solution | Maintains integrity of the product. No protein structural damage. | [43] |
| Sea bass (*Dicentrarchuslabrax*) | PSF −15 and −25 °C, 200 MPa | Reduction of ice crystals, lower protein denaturation and higher water holding capacity, compared to traditional freezing techniques. | [44] |
| Large yellow croaker (*Larimichthyscrocea*) | UAF 20, 28 and 40 kHz, 175 W, 30s | Increased freezing rate, preserved quality parameters and reduced lipid oxidation. | [45] |
| Salmon carpaccio | Gelatin-Ch films | Moderate antimicrobial activity. | [46] |
| White shrimp (*Penaeus vannamei*) | Microalgal exopolysaccharide coating | Inhibition of microbial growth and deterioration. Preservation of sensory properties. | [47] |
| Smoked herring (*Clupea harengus*) | Ch coating | Inhibition of microbial growth and lipid oxidation. Antioxidant effects. Improvement of sensory parameters. | [48] |
| Pacific white shrimp (*Litopenaeusvannamei*) | Oregano essential oil and ε-polylysine applied in the surface | Inhibition of microbial growth and proteolysis. Improvement of sensory properties. | [49] |
| Asian sea bass (*Lates calcarifer*) | WPI coating with ginger, green tea, and lemongrass polyphenol extracts | Inhibition of microbial growth, proteolysis, lipid oxidation and other deterioration markers. Reduction of muscle softening and color and drip loss. | [50] |
| Black tiger shrimp (*Penaeus monodon*) | Ch-gelatin edible coating incorporated with longkong extract | Inhibition of microbial growth, proteolysis, and lipid oxidation, melanosis, and other deterioration markers. | [51] |
| Yellow croaker (*Larimichthyscrocea*) | Ch -lysozyme edible coating | Inhibition of microbial growth, proteolysis, and lipid oxidation. Color and odor characteristics were preserved. | [52] |
| *Scomberoidescommersonnianus* | Ch-WPI-*Artemisia dracunculus* essential oil coating | Inhibition of microbial growth, proteolysis, and lipid oxidation. Preservation of sensory properties (color, odor, texture, and appearance) during storage. | [53] |
| Hairtail (*Trichiurushaumela*) | Eugenol- Ch nanoemulsion | Inhibition of microbial growth, proteolysis, and lipid oxidation. Improvement of sensory scores. | [54] |
| Beluga sturgeon (*Huso huso*) | Jujube gum nanoemulsions containing nettle essential oil edible coating | Inhibition of microbial growth, lipid oxidation and loss of sensory properties. | [55] |
| Cod (*Gadus morhua*) | Nano-ZnO into packaging material | Inhibition of microbial growth, reduction of water loss, gumminess, and adhesiveness. | [56] |
| Tiger tooth croaker (*Otolithesruber*) | Poly lactic acid film containing ZnO nanoparticles and essential oils | Inhibition of microbial growth, antioxidant properties. Increase of shelf life during storage. | [57] |

Abbreviations: ICF, isochoric freezing; PSF, pressure shift freezing; HPF: high pressure assisted freezing; UAF, ultrasound assisted freezing; Ch: chitosan; WPI, whey protein isolate.

*3.1. Freezing-Based Technologies*

Freezing has been extensively used in industry to preserve different food products, including seafood. However, some methods such as air blast freezing or cryogenic freezing could reduce the quality of the product, due to the slow freezing rate, forming large, irregular, and not evenly distributed ice crystals. This can lead to damage of muscle

tissues, protein denaturalization and changes in texture, water-holding capacity, and color of the food product [41,58]. To solve these drawbacks, novel freezing methods such as pressure-related freezing techniques (PF), ultrasound assisted freezing (UAF), electrically assisted freezing, magnetically assisted freezing, or the use of anti-freeze proteins have been designed, benefiting from the advantages of implementing of 4IR technologies [41]. However, few studies have been performed in seafood field. Therefore, it can be expected that in the coming years more scientific studies will evaluate the usefulness of these technologies in these foods.

### 3.2. Edible Films and Coatings

In the recent years, edible films and coatings are being increasingly studied due to various advantages over synthetic materials used for food packaging, such as polyethylene, polyamide, and polypropylene. These advantages include efficacy to retard food degradation and extent shelf life, no toxicity, and environmental friendliness [48,59]. Development of edible films and coatings has been powered by recent advances of 4IR technologies and related fields, especially nanotechnology. Sometimes, no distinction is made between films and coatings, but they have different functions and are applied in different ways. Generally, edible films are prepared separately as solid sheets and then used to cover the surface of the food, whereas coatings are formed directly onto the food surfaces [59,60]. Most of edible materials are formed from natural biopolymers of polysaccharides (e.g., cellulose, starch, alginates, chitosan, etc.) or proteins (e.g., casein, whey, wheat gluten, etc.). In recent years, different studies have tested the efficacy of these materials in different seafood products, as shown in Table 2. For example, a study evaluated the efficacy of a gelatin-chitosan edible film to preserve salmon carpaccio [46]. The results showed that the material had moderate antimicrobial activity against several foodborne pathogens and enhanced the shelf-life of the product. Similarly, different chitosan coatings were reported to have antimicrobial and antioxidant properties and also enhanced the physicochemical and sensory properties of smoked herring during cold storage [48].

### 3.3. Natural Preservatives

Among preservation techniques, the use of natural preservatives to extended the shelf-life of seafood products has gained a great attention in recent years, mainly due to the current consumer's demands for more natural strategies to substitute synthetic additives [42,61]. In this sense, natural additives with antimicrobial, antioxidant, and anti-browning activity have been tested for their ability to prevent contamination and loss of organoleptic properties of seafood products. These compounds can be obtained from vegetal, animal, or microbial sources, being some of the most utilized essential oils, plant extracts, and chitosan [61–63]. These natural additives can be used directly to the seafood product, but they can be also added into the packaging material, especially into edible films. Some of the most recent studies using natural preservatives in seafood products have been compiled in Table 2, which have been tested in different seafood products. Several natural additives from vegetal origin, different plants like oregano, thyme, rosemary, grape, clove, or tea have been successfully used to preserve seafood. Essential oils and compounds like terpenes, alkaloids and phenolic compounds present in plant extracts are usually the compounds involved in the antioxidant and antimicrobial activities [61–63]. For example, recently, the antimicrobial activity of a combined treatment of oregano essential oil and $\varepsilon$-polylysine was tested in Pacific white shrimp samples [49]. The in situ experiment demonstrated that the treatment significantly reduced the growth of *Shewanellaputrefaciens* and proteolysisand also improved the sensoryproperties of the shrimps (better texture, cephalothorax color, and lower drip loss).

### 3.4. Nanotechnology

Nanotechnology field has been broadly recognized as one of the most innovative fields in the 21st century. The small size and greater surface area lead to unique and novel

properties that can be used in many areas, including in food industry. In this field, nanotechnology has been used in different applications, including food production, processing, preservation, and packaging [64,65]. Regarding food preservation, these applications include different aspects (Table 2). First, the use of nanostructures may be helpful to improve the accessibility of low soluble compounds, bioactivity, stability, or control their release into the product by nanoencapsulation or nanoemulsions [54,55,64]. For example, a recent study elaborated a eugenol-chitosan emulsion to improve the solubility of the former and reduce its evaporation while improving the antioxidant and antimicrobial properties of chitosan. The material successfully reduced microbial growth and improved the sensory properties of hairtail during storage [54]. Second, nanocomposites can be employed to reduce microbial growth and maintain the freshness of the products. Diverse nanocomposites such as silver, zinc oxide, titanium oxide, or chitosan nanoparticles have been employed in food preservation [64,65]. Just to cite some recent examples, zinc oxide nanoparticles were included into packaging material and tested in the preservation of cod and tiger tooth croaker, showing antimicrobial activity and the ability to increase their shelf life [56,57]. Other examples of the application of nanotechnology in the preservation of seafood can be found in Table 2.

## 4. Emerging Trends in Seafood Processing Methods

### 4.1. Thermal Processing

Extensively employed in the industrial processing of seafood, thermal processing is approved to prevent pathogenic and spoilage bacteria growth and extend shelf-life of seafood (Table A1). Traditional thermal processing treatments (e.g., boiling, steaming, roasting, and deep-frying) have been applied for a long time, although they are often associated with considerable challenges, such as long processing time and negative impact on sensory properties and nutritional quality. Recently, with the emergence of the 4IR, innovative heating technologies (e.g., microwave and radio frequency heating, ohmic heating, and infrared heating) have gained increased industrial interests. Ohmic heating, dielectric heating (i.e., radio frequency and microwave heating), and infrared heating are promising alternatives to conventional methods of heat processing. These novel thermal technologies are viewed as volumetric heating (internal heating) whereas conventional heating methods are based on convection and conduction (external heating) [66,67].

Due to its desirable features, ohmic heating has found a large number of applications such as heating, cooking, thawing, blanching, evaporation, dehydration, pasteurization, fermentation, and extraction [68,69]. The benefits related to ohmic heating are numerous compared to traditional heating techniques. The technique achieves high heating rates and more uniform heating compared to conventional heat treatments. In addition, this technique is environmentally friendly and enables high energy efficiency [67–70]. Several potential applications of ohmic heating on seafood products can be found in the literature. For example, the effect of various ohmic heating treatments on the textural properties of Pacific whiting and Alaska pollock surimi gels, mixed with carrot cubes was assessed by Hoon Moon and co-authors [71]. The combined impact of ohmic heating and high pressure on the thermal and structural properties of shrimp was also studied by Dang and others in a recent study [72]. In another study, the ohmic heating was found to decrease the thawing time, thawing loss, and total loss of frozen tuna fish cubes compared to conventional heating [73].

Microwave has become a popular method that converts electromagnetic energy into thermal energy, and has several applications (such as sterilization, cooking, extraction, and thawing) in the food industry [74]. Microwave heating has various desirable characteristics such as high heating rates, ease of use, safe handling, and low maintenance. Moreover, the technique causes less damage to sensory and nutritional quality of food compared to conventional heating [75,76]. The superiority of microwave heating over conventional methods, such as conventional pasteurization, oven, water bath has been demonstrated in several studies carried out on a wide range of seafood products. For example, a recent study

showed that microwave heating improves the gelling properties of surimi gel fortified with fish oil and boosts the chemical forces involved in the gel formation [77]. Another study investigated the impact of microwave heating on the gel characteristics of surimi, and the results indicated that thickness of the surimi was the most important factor that influences the temperature distribution during microwave heating [78]. However, the cost of the equipment hinders its wider deployment at large-scale facilities [66,79].

More adoption of Industry 4.0 technologies is expected in the future, enabling seafood processing and preservation to address the current few limitations associated with emerging thermal treatments techniques (e.g., ohmic and microwave heating).

### 4.2. Nonthermal Processing

The use of nonthermal treatments has been extensively studied in recent years to meet consumer demand for minimally processed products with high sensory and nutritional qualities. Nonthermal pulsed electric field (PEF), high pressure processing (HPP), and ultrasound (US) are among the most applied techniques [80,81].

PEF processing has the potential to extend food shelf-life, maintaining high nutritional value and acceptable sensory properties [82]. PEF processing involves using short-duration pulses (1–100 s) in strong electric fields (0.3–4 kV/cm). The use of PEF in various applications in seafood treatments and preservation has lately surged in popularity. However, PEF-treated seafood samples could have greater oxidation samples of primary and secondary oxidation products and protein carbonyls than untreated fish because of damage to cell membranes that occurred during electroporation [83]. PEF-induced alterations in the texture and sensory qualities of seafood may have both beneficial (e.g., better tenderization) and negative (e.g., reduced functional properties, decreased nutritional quality and safety) effects. The application of PEF processing in the seafood industry are currently limited, despite the technique's efficiency and eco-friendliness [21]. Some applications of PEF in extraction of by-products, preservation, and pre-treatment of seafood can be found in the literature (Table 3).

**Table 3.** Examples of applications of PEF for extraction purposes in seafood.

| Sample Food | PEF Protocol | Extracts | Findings | Reference |
|---|---|---|---|---|
| Fishbone | 20 kV/cm; 8 pulses, 120 min | Calcium | 77.1% extraction efficiency | [84] |
| Fishbone | 22.79 kV/cm, 9 pulses, 2 s | Calcium and chondroitin sulfate | Extraction is much quicker and contains significantly more CS than standard approaches | [85] |
| Mussel | 20 kV/cm, 8 pulses, 2 h | Protein | Extraction is much quicker and increased extraction (77.08%) | [86] |
| Haliotis discus hannaiInoviscera | 20 kV/cm, 600 s | Protein hydrolysate | an improved yield of abalone viscera protein | [87] |
| Fish residues | 1.4 kV/cm, 20 s, 10 Hz, 100 pulses | Antioxidant extraction | Enhanced the antioxidant capacity | [88] |
| Residues of rainbow trout and sole | 1–3 kV/cm, 123–300 kJ/kg, 15–24 h | Protein | 80% enhanced | [89] |

Fishery by-products are abundant in physiologically active molecules, thus using environment-friendly and effective technologies to extract the high-value compounds is critical. Fish bones, gills, and heads are the principal fish leftovers, comprising 30–70% of a total weight. The PEF approach may also be utilized to valorize these by-products. PEF has been proposed as an ecologically acceptable and cost-effective approach for extracting antioxidant-active extracts from the fish processing industry [88,90]. For example, Franco and others advised using PEF treatment to improve antioxidant extraction from fish residues for two commercial fish species (sea bream and sea bass) [88]. PEF (1.4 kV/cm, 20 s, 10 Hz, 100 pulses) enhanced the antioxidant capacity compared to water or methanol extracts.

Shiekh and Benjakul investigated the role of PEF in microbiological alterations in Pacific white shrimp, a widely consumed invertebrate [91]. The shrimps were subjected to PEF treatment at various energy densities (54–483 kJ/kg) and pulses (200–600) and then stored at 4 °C for 10 days. The authors found PEF might be a useful technique for inhibiting the psychrophilic bacteria that cause shrimp deterioration during the storage at low temperatures [92]. In another study, the impacts of PEF (1–2 kV/cm, 20 s) in combination with carbon-dioxide ($CO_2$: 70%) and high-hydrostatic pressure (150 MPa, 5 min) were investigated on the physicochemical properties and shelf life of *coho* salmon during 25 days of refrigerated storage [93]. Pre- and post-rigor fish texture changes were observed to be different between untreated and treated salmon, as improved hardness and chewiness values were observed for pre-rigor PEF processed salmon. Additionally, the combined technologies significantly decreased the protease and lipase activities of pre-rigor fish.

As a pre-treatment, PEF technology was recommended as a feasible technique to improve water retention capacity [94]. It was also reported that PEF treatment could result in lower peroxide and TBARS levels [83]. Sea bass samples were pre-treated with PEF before brine-salting to validate the possibility of accelerating the brining rate, enhancing salt absorption, and assuring uniform salt distribution throughout the muscle [95]. The current was delivered at 10 and 20 A (0.3–0.6 kV/cm) before sea bass salting in 5 and 10% salt brine, respectively. PEF pre-treatment reduced the brine salting time by 5 days compared to control samples and increased salt absorption by 77% while maintaining homogeneous distribution. The aforementioned literature studies confirm that PDF has a huge potential to contribute to more sustainable and robust seafood production in the age of Industry 4.0.

In addition to PEF, other innovative processing techniques have been widely used in recent years. HPP is one of the novel nonthermal processing techniques that have growing industrial applications. In seafood, the technique is basically used for killing microorganisms and parasites, extending shelf life, pressure shift freezing or thawing in addition to deboning procedure of crustaceans and bivalves for thorough detachment of meat [96]. The capability of HPP to inactivate microorganisms and enzymes depends on pressure level, pressure holding time/temperature and product characteristics. For inactivating microorganisms and maintaining quality, food is exposed to pressure levels ranging between 200–600 MPa and holding times between 2–10 min [97]. A wide range of seafood products, e.g., fresh, or smoked salmon, red mullet, rainbow trout, mackerel and bivalve shellfish, such as lobster, and oyster can be treated by using HPP. Examples of these treatments are shown in Table A2.

Ultrasound (US) is another innovative emerging non-thermal food processing technology that is based on sound waves having frequency (20 kHz) higher than human hearing ability. The technique is extensively used in food processing procedures including, preservation, texture tenderization, extraction, emulsification, freezing, thawing and microbial inactivation [98,99]. On the basis of used frequency value of ultrasound, it can be divided into two groups as low energy group with frequency ranging from 5–10 MHz with intensity of <1 W/cm$^2$ and high energy group with frequency ranging from 20–100 kHz and intensity of >1 W/cm$^2$. The group of low energy ultrasound is used for the non-destructive examination of food quality, while the group of high energy ultrasound is disrupting, and can produce physical and chemical changes in food properties, thus killing microbes while maintaining food quality [100,101]. US treatment is used for inactivation of microorganisms due to the production of cavitation bubbles, shear disturbance, creation of microstreaming and shear force, and formation of free radicals [102,103]. These conditions destroy the cell wall of microorganisms which leads to their decay [103]. The technique can be used alone or in combination with other techniques (based on heat or pressure) on various fish products to increase safety of microbiological and sensory qualities [80,104,105]. Recently, an innovative US-based technology has been emerged and used for the extraction of bioactive compounds known as ultrasounds-assisted extraction (UAE). This technique

improves both extraction time and extracted yield, and also enhances working efficiency by reducing the particle size. Presently, ultrasound-assisted extraction is broadly used for the extraction of many valuable bioactive compounds from seafood by-products [106]. Examples of application of US in the seafood industry can be found in (Table A3).

From the aforementioned examples, it could be concluded that HPP and US would improve seafood processing and industrial manufacturing and save time and energy thanks to automation and digitalization afforded by food industry 4.0 technologies.

## 5. Recent Advances in Seafood Analytical Methodology

### 5.1. Hyperspectral Sensors

Spectroscopy and spectral imaging techniques have been given special attention in recent years being fast and non-destructive fingerprinting methods. The potential of several spectroscopic techniques operating in the ultraviolet (UV), visible (VIS), near infrared (NIR) or mid-infrared (MIR) regions has been widely investigated, with the VIS-NIR regions being the most used analysis for fish and other seafood products. Currently, some of these techniques are being moved from the laboratory into real-world applications in industry to manage different issues, such as determination of chemical composition, color, and other physical properties, microplastic detection, microbial spoilage, authenticity issues, as well as process monitoring. Smart sensors based on hyperspectral imaging (HSI) have received enormous interests. The increased attention of the seafood industry towards HSI based devices is due to its properties to generate automatically, rapidly, and in one shot spatial and spectral information without the need for solvents or highly trained staff. Therefore, HSI can be considered as a promising building block for the Industry 4.0 (Figure 1).

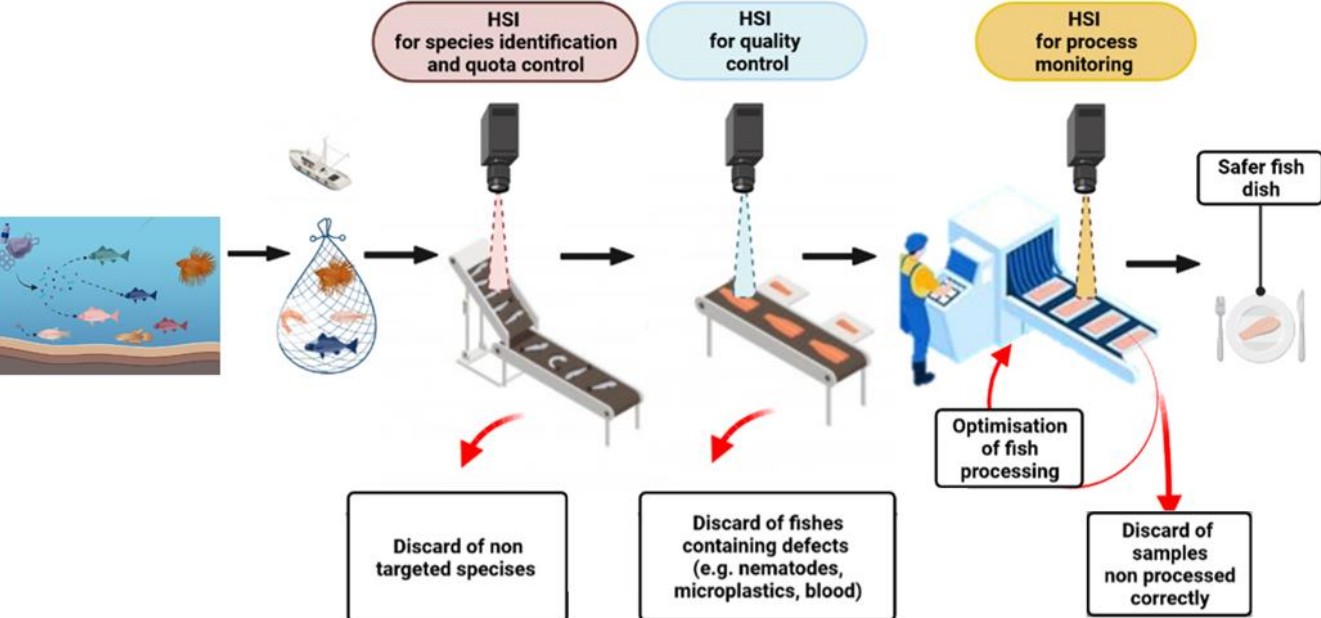

**Figure 1.** Examples of applications of hyperspectral imaging in the seafood sector.

### 5.1.1. Chemical Properties

HSI has been mostly used for prediction of parameters related to fish freshness (e.g., K-value and total volatile basic nitrogen; TVB-N) or basic chemical composition (moisture content, protein, fat, or fat-related substances). For example, Cheng et al. [107] used Vis-NIR HSI (308–1105 nm) to predict K-value in grass and silver carps and obtained good results for this prediction, with a coefficient of determination for prediction ($R^2p$) equal to 0.94. In another study, Cheng et al. [108] succeeded in increasing the prediction accuracy of TVB-N ($R^2p = 0.98$) by combining HSI and image texture properties based on gradient co-occurrence matrix difference and an innovative algorithm, called physarum

network. This was confirmed by Wang et al. [109] using NIR HSI of fish fillet and on HSI and textural features images of fish eyes and gills.

### 5.1.2. Color and Other Physical Properties

The detection and quantification of blood in the fish fillet is one of the most important quality factors for the white fish industry because of its impact on consumer's preference. In this context, Skjelvareid et al. [110] successfully used VIS-NIR HIS imaging based on diffuse reflectance measurement to detect and quantify blood in cod fillets. Texture, as an important factor for fish quality evaluation, was also studied using HSI technique. Ma et al. [111] used HSI in the VIS-NIR region to evaluate different textural (Warner–Bratzler shear force hardness, gumminess, and chewiness) parameters of vacuum freeze-dried fish fillets. The authors reported high $R^2p$ values (>0.84) for all the investigated parameters.

### 5.1.3. Microplastic Evaluation

Microplastics in marine ecosystems are those small plastic fragments that can be consumed by aquatic species (e.g., fish, shellfish, marine invertebrates) and transferred along the food chain to human beings [112–115]. To detect and identify microplastics, several approaches have been developed, ranging from the simple visual inspection to more advanced techniques, such as chromatography coupled with mass spectrometry and spectroscopic techniques [114,116–119]. A recent application with HSI was proposed by Zhang et al. [120] to evaluate in-line microplastic contamination (quantification and identification) of *Carassius carassius*. The sample analysis was performed without prior separation from the intestinal tract and proved that five microplastic types (polyethylene, polystyrene, polyethylene terephthalate, polypropylene, and polycarbonate) can be identified in only 36 min with a good accuracy (classification factors >96.22% for recall and precision). This study paved the way for future industrial applications in fishery industry in order to provide safer fishery products to consumer.

### 5.1.4. Microbial Spoilage

The prediction of harmful microorganisms directly in fishery products is vital for the food industry in order to ensure quality and safety to consumers. In the past five years, limited studies for evaluation of microbial quality in fishery products were reported. For example, HSI has been used for evaluating different microorganisms (TVC, *Escherichiacoli*, *Pseudomonas*, and *Enterobacteriaceae*) in fishes [121–124]. In these studies, VIS–NIR HSI technique was applied with different wavelength ranges for HSI acquisition (400–1000 nm and 900–1700 nm). Depending of the microorganism considered, the prediction accuracy varied between $0.80 < R^2p < 0.96$. Recently, Khoshnoudi-Nia et al. [125] confirmed the HSI suitability for psychotropic plate count prediction ($R^2p = 0.921$) in rainbow trout fillets.

### 5.1.5. Authentication

In seafood industry two fraudulent practices are commonly encountered; the substitution of high-priced fish species with inexpensive ones and mislabeling fresh/frozen-thawed. Cheng et al. [126] used successfully Vis-NIR-HSI to distinguish fresh from cold-stored (4 °C for 7 days) and frozen-thawed (−20 °C and −40 °C for 30 days) grass carp fish fillets. Qu et al. [127] confirmed the previous results on shelled shrimp (*Metapenaeus ensis*). The authors demonstrated that fresh shrimps and those from either cold storage or freezing could be distinguished with accuracy more than 88%. Later, Xu et al. [128] demonstrated that HSI can be used to discriminate fish in fresh and chill-stored conditions based on their farming system (organic vs. conventional) at a rate of 98.2%. Recently, Qin et al. [129] studied the potential of three HSI systems (Vis-NIR, fluorescence, short-wave infrared, and Raman) to discriminate between fish fillets of six species (i.e., Red snapper, Vermilion snapper, Malabar snapper, Summer flounder, White bass, and Tilapia). The best classification was provided by Vis-NIR technique, giving 99.9% of correct classification rates.

### 5.1.6. Process Monitoring

One of the main benefits of HSI is its ability to be applied on production lines, making it a promising choice for real time measurements, in-line process monitoring, and optimization of key process parameters. For example, a HSI (900–2500 nm) system was used to classify a Japanese seafood product (called Kamaboko) as a function of their thermal treatments [130]. Core temperature and thermal history of that product were predicted with good precision by both partial least squares regression model and linear discriminant analysis. The results showed that HSI can be used to visualize if the seafood has reached the targeted thermal temperature. Recently, a point measurement NIR system (760–1040 nm) and a commercial in-line NIR imaging scanner (QMonitor, TOMRA, Asker, Norway) were successfully used for determining the core temperature of heat-treated fish cakes [131]. HSI based on fluorescence measurement was also used [132] and proved its efficiency to monitor cooking temperature in cod (*Gadus morhua*) fillets processed at different temperatures (30, 50, and 70 °C).

### 5.2. Advanced Mass Spectrometry and Chromatography

Nowadays, and transversal to several fields of science—including medicine, chemistry, pharmacy, and other related fields, high resolution chromatographic techniques (HRCT) and mass spectrometry (MS), has evolved as one of the most valuable analytical tools available to modern scientists allowing a deep and comprehensive knowledge on structural elucidation of unknown substances [133], quality control of drugs [134], clinical [135], environmental [136], food control [137], and forensic analytes [138], providing qualitative as well as quantitative information for a broad variety of compound classes. Other applications include inorganic chemical analysis, geochronology [139], reaction kinetics [140], determination of thermodynamic parameters and ion-molecule reactions [141]. Unequalled sensitivity, very low detection limits, speed and diversity of its applications are analytical characteristics that have raised MS and HRCT to an outstanding level among analytical methods. Generally, HRCT acts as the inlet system of most of MS instruments and therefore will allow very high-resolution power and extreme selectivity intended to improve the performance of MS. Currently, the output of HRCT and MS has reached an unprecedented level thanks to highly automated systems offered by the Industry 4.0 technologies, very efficient and powerful ionization methods, and the possibility of combination of MS in various ways.

The most important milestones which boosted significant advancements of MS are highlighted in Figure 2. These advancements have been empowered by the development of different ionization methods (electrospray ionization; ESI, matrix-assisted laser desorption/ionization; MALDI) and the combination of HRCT with MS, allowing MS to become a powerful technique in several research fields.

These revolutionary advancements allowed the scientists to have a much deeper and comprehensive understanding into molecular world. Moreover, recent technological innovations in MS-based techniques have contributed to a range of highly sensitive and versatile instruments for high-throughput, high-sensitive, and OMIC-scale profiling.

Since the development of ESI and MALDI techniques the application of MS in OMIC fields has increased exponentially. Direct infusion-mass spectrometry (DI-MS), mass spectrometry imaging (MSI), direct analysis in real time MS, and MALDI-MS experienced significant advances in addition to separation-based MS techniques such as ion mobility-mass spectrometry (IM-MS), liquid chromatography-mass spectrometry (LC-MS), gas chromatography-mass spectrometry (GC-MS), and capillary electrophoresis-mass spectrometry (CE-MS). In a recent study, Freitas et al. [142] explored, for the first time, the possibility to differentiate between *Sparus aurata* from two different mariculture farms using the mass fingerprint of fish mucus obtained from MALDI-TOF MS. Freitas et al. [143] also developed a systematic analytical quality-by-design approach combined with GC-MS for the quantification of volatile amines trimethylamine and dimethylamine in *Sparus aurata*,

as important spoilage indices for seafood products, assisting in the determination of the rejection period.

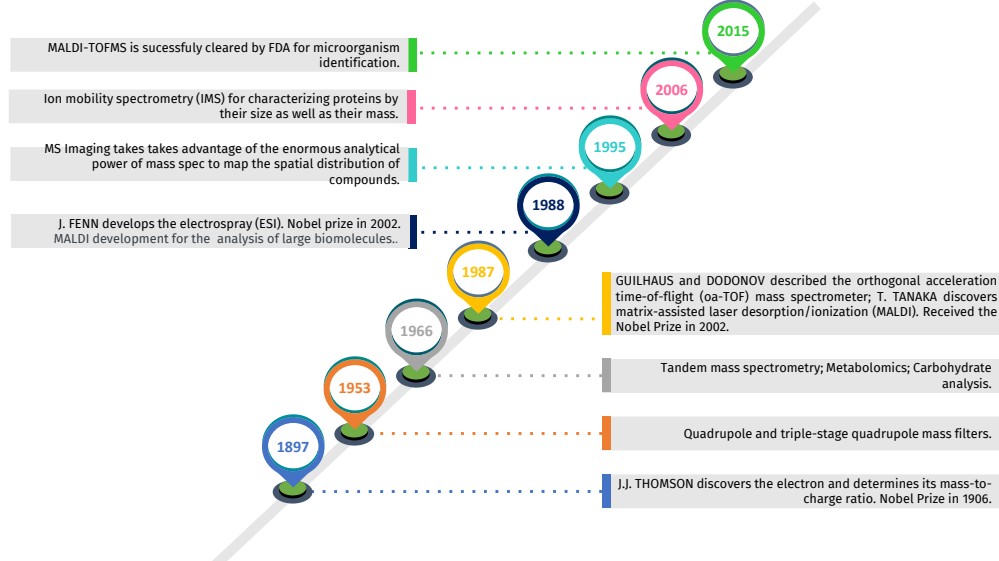

**Figure 2.** Some important milestones responsible for the important advances in MS.

Currently, in addition to chemical information analysis of the extract content by GC-MS, CE-MS, LC-MS or DI-MS methods, MS technologies allow the use of imaging methods to analyze intact tissue or cells providing useful information for the location of given metabolites. Indeed, imaging MS techniques allow obtaining the spatial distribution (such as detection the presence or absence of certain proteins and if they are collocated with other proteins and their location in histologically well-defined areas) of a large number of intact protein and biomarkers [144]. MS has progressed very quickly during the 4IR era, leading to the advent of new and improved instruments (Figure 3). The rapid advances in technology and instrumentation associated to spectacular advancements in OMICs field have driven by the methodological and technological improvements of HRCT and MS techniques.

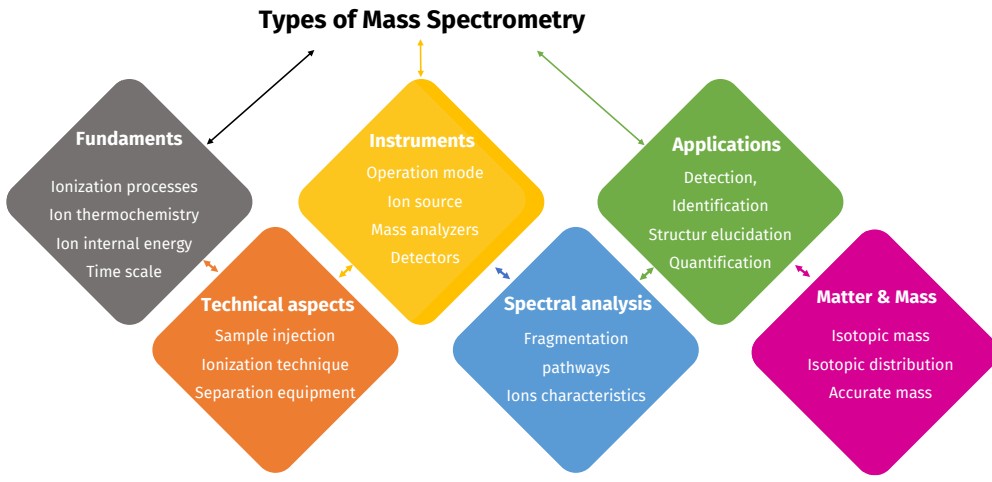

**Figure 3.** Recent technical innovations in MS-based techniques.

Numerous studies dealing with MS techniques can be found in the literature for different seafood applications. For example, a direct analysis in real time (DART) coupled to high resolution mass spectrometry (HRMS) was developed to differentiate farmed from wild salmon from Canada, Chile, and Norway [145]. Principal component analysis applied to most abundant signals generated from fatty acids after the DART-HRMS analysis of

sample lipid extracts, showed a clear separation between farmed and wild fish. Recently, the same technique was used to discriminate between fresh and frozen-thawed sea bass (*Dicentrarchuslabrax*) by concatenating a low-level data fusion strategy to multivariate statistical analysis [146]. The score plot of partial least squared discriminant analysis applied to DART-HRMS data showed a perfect discrimination between fresh and frozen-thawed fish.

A support vector machine (SVM) classification model was then built with 25 molecular features and the developed model was then validated on independent sets of sea bass and on salmon. The same research team identified, in another study using LC-HRMS/MS, two markers, namely EPA and DHA of freezing-thawing [147]. These fatty acids were used to build a model capable of classifying European sea bass samples according to their fresh or frozen/thawed condition. Therefore, the work highlights the importance of the chromatographic and spectrometric techniques on quality and safety food parameters, opening the door for the development of a targeted method providing robust species-independent tests to be applied in the seafood chain by producers and competent authorities. In another study, MALDI-MSI was used with success to identify two biochemical indices (formaldehyde derived from the decomposition of trimethylamine N-oxide and protease-induced softening) that could be used to evaluate the quality of *Macruronus novaezelandiae* fillets [148]. Recently, Shen and co-authors [149] authenticated successfully cod (most valuable fish species) and oil fish (less expensive fish species) using rapid evaporative ionization MS coupled to intelligent surgical knife.

### 5.3. Other Advanced Techniques

In addition to the aforementioned MS techniques and chromatographic methods, and HSI, other conventional and emerging analytical approaches have been widely applied to evaluate quality, authenticity, and adulteration of fish and other food. For example, Fourier transform infrared (FTIR), Raman, nuclear magnetic resonance (NMR), immunological assays such as enzyme-linked immunosorbent assay (ELISA), and DNA-based techniques are among the most used analytical strategies. These techniques are not thoroughly detailed in this review paper as they are well covered by other recent published review papers. For example, Hong-Ju et al. [150] reviewed research advances of non-destructive optical techniques in the application of fish quality evaluation. A similar overview of spectroscopic techniques and other conventional and emerging methods was also given in other papers [27,151,152]. Optical array sensing technologies, focusing on colorimetric and fluorometric sensor arrays, and electronic noses and tongues, and their applications in, among others, quality control of foods and beverages were also reviewed in a comprehensive literature analysis [153].

Several review papers highlighted recent technological advances in DNA-based identification methods, with a focus on seafood species identification and origin in automated and high-throughput settings. Polymerase chain reaction (PCR) techniques such as PCR-restriction fragment length polymorphism, species-specific PCR, real-time PCR, and multiplex PCR are among the most used techniques [154,155]. El Sheikha and Montet [156] suggested to combine PCR with Denaturing Gradient Gel Electrophoresis (PCR-DGGE) to discriminate the geographical origin of fish. The current status on the application of DNA barcodes to seafood species authentication, through the description of the barcode regions, reference databases, and related methodologies was reported by Telmo et al. [155].It should be highlighted that, the recent advances in data processing, analysis, and presentation, as well as data analytical techniques (AI, ML, and big data analytics), and the combination of these analytical techniques with chemometrics makes them more convenient and effective for the analysis of a wide variety of foods in the age of Industry 4.0.

## 6. Perspectives and Future Directions

There is ample evidence that the technological advancements discussed in the sections above have improved current seafood processing and preservation methods, especially in

the land-based seafood industry, which delivers excellent, high quality products with good shelf-life. Thanks to the ability to distribute thermal energy homogeneously within the muscle matrix, innovative thermal technologies such as ohmic, microwave, dielectric, and infrared heating are among the promising alternatives to conventional methods, especially for the development of new ready-to-eat seafood products. These techniques also have the advantage of being environmentally friendly and energy efficient. They are therefore aligned with the United Nations Sustainable Development Goals (SDGs) that will shape world development plans over the next decade and are of great interest to Industry 4.0.

As for non-thermal technologies, the use of PEF treatment and ultrasound technology to improve the quality and shelf-life of seafood products has advantages (e.g., optimal control of the microbial population) and limitations (e.g., sensory changes); they are nonetheless very promising with respect to the extraction and purification of bioactive compounds from fishery discards (i.e., non-target species that are returned to the sea) and/or industrial seafood waste (i.e., skin, head, shells, backbones, etc.). The great progress in the field of smart and biodegradable packaging, such as the new eco-friendly edible films and coatings, natural preservatives, and nanotechnology, may soon help address one of the main challenges of the 21st century, namely, to reduce consumption of plastic bags. In addition, seafood waste as a rich source of bioactive compounds could soon realize the dream of transforming food waste into an added-value resource for the production of new sustainable packaging materials. This review also provides basic information on seafood authentication and surveillance. This is an area of growing concern, given that about 30% of seafood products are subject to mislabeling. Some rapid spectroscopic techniques, such as HSI, Vis-NIR, fluorescence, short-wave infrared and Raman, as well as omic technologies, such as microbiome profiling, proteomics, and metabolomics, could offer promising alternatives to conventional approaches in discriminating between species and between fresh and frozen-thawed products.

Despite these advances, several innovative technologies reported in this review are difficult to implement onboard fishing vessels (where post-mortem changes in most fishery products begin) and in developing countries, where a significant fraction of worldwide seafood products are produced despite the lack of adequate operating conditions and basic infrastructures (including about 1.7 million non-motorized fishing vessels that represent 40% of the total world fleet). In fact, some emerging technologies such as fast freezing (based on cryogenic fluids, high pressure, or ultrasound) and HPP are difficult to install and maintain onboard offshore fishing vessels. This is even more the case in many small-to-medium scale fisheries in Africa, Asia, and South and Latin America, where even common flake ice and refrigerators both onboard and in the landing areas (or in the vicinity) are still a precious commodity. Consequently, millions of tons of landed fishery products (e.g., fish, cephalopods and crustaceans), most likely destined for international markets, are seriously compromised or spoiled (from both a chemical and a microbiological perspective) even before they reach the collection centers.

From this perspective, Industry 4.0 can help to promote the use of more affordable equipment among fishers and fish farmers, improving workplace conditions as well as seafood hygiene and quality. Industry 4.0 technologies can also play a crucial role in supporting a more sustainable future, for example by maximizing the value of marine resources, thus maintaining world fish stock within biologically sustainable levels. Lastly, from a social perspective, the fourth industrial revolution can help improve health and well-being in developing countries, foresting wider access to broader education, information and greater economic opportunity [157]. In addition, broader collaborative efforts among researchers, industry and international institutions are required to maximize the benefits of Industry 4.0 and overcome the above paradoxical inequalities between land-based industry and fishing vessels and between developed countries and the rest of the world. For instance, the development of new intelligent, flexible, agile production platforms capable of better connecting supply and demand and of adapting quickly to consumer needs would allow a drastic reduction in production inefficiency, waste, emissions, and other costs.

To conclude, although Industry 4.0 has brought direct benefits to advanced seafood companies, as reported in this review, it should broaden its perspective to successfully address three important components of a global problem: The growing demand for safe, high-quality fishery and aquaculture products, the valorization of industrial seafood waste and marine species that are discarded into the sea due to their low commercial value, and, lastly, the increasingly high levels of poverty and malnutrition worldwide. In line with the key issues above, in the next 10 years seafood scientists and 4.0 industries must address three ambitious goals set in 2015 by the UN in the abovementioned SDGs agenda: Good Health and Well-Being, Responsible Consumption and Production, Zero Hunger. This paper reviewed relevant studies dealing with development and technologies of 4IR that are associated with processing/preservation and analytical techniques in the seafood sector. It would be interesting to review in future work the role of Industry 4.0 technologies in automation and digitalization of meat production and other food industries.

**Author Contributions:** Conceptualization, methodology, writing—original draft preparation, A.H.; writing—original draft preparation, S.A.S., S.S., İ.U., R.N.A., P.G.-O., M.A.P., A.A.-K., R.P., J.S.C., and G.B. All authors have read and agreed to the published version of the manuscript.

**Funding:** This research received no external funding.

**Institutional Review Board Statement:** Not applicable.

**Informed Consent Statement:** Not applicable.

**Acknowledgments:** José S. Câmara and Rosa Perestrelo acknowledge FCT-Fundação para a Ciência e a Tecnologia through the CQM Base Fund—UIDB/00674/2020, and Programmatic Fund—UIDP/00674/2020, Madeira 14–20 Program, project PROEQUIPRAM—Reforço do Investimento em Equipamentos e Infraestruturas Científicas na RAM (M1420-01-0145-FEDER-000008), and ARDITI—Agência Regional para o Desenvolvimento da Investigação Tecnologia e Inovação, through M1420-01-0145-FEDER-000005—Centro de Química da Madeira—CQM+ (Madeira 14–20 Program) for their support. The research leading to these results was supported by MICINN supporting the Ramón y Cajal grant for M.A. Prieto (RYC-2017-22891); by Xunta de Galicia for supporting the program EXCELENCIA-ED431F 2020/12; and the pre-doctoral grant of P. Garcia-Oliveira (ED481A-2019/295); and by the program BENEFICIOS DO CONSUMO DAS ESPECIES TINTORERA-(CO-0019-2021).

**Conflicts of Interest:** The authors declare no conflict of interest.

## Appendix A

**Table A1.** Use of thermal processing methods in seafood products.

| Thermal Processing | Product | Treatment Purpose | Process Condition | Objective | Main Findings | Reference |
|---|---|---|---|---|---|---|
| Ohmic | Pacific whiting and Alaska pollock surimi gels prepared with carrot | Heating | Heating from 5 to 90 °C under 3 rates: 3.3 V/cm (3 °C/min), 12.0 V/cm (60 °C/min), and 17.3 V/cm (160 °C/min) | Evaluate the textural properties of surimi and vegetables during surimi gelation | Shear force value of carrot heated at 160 °C/min showed the highest value in the both species. Moisture loss (%) of surimi and carrots heated at 160 °C/min was significantly lower than other samples heated slowly. | [71] |
| Ohmic and high pressure heating | Shrimp | Thawing | -Ohmic heating: Voltage; 92, 138, and 184 V, 60 Hz, 40 A. T°: 30, 35, 40, 50, and 60 °C -High pressure heating: 100 and 600 MPa for 30 s–3 min at 5–20 °C | Study the peelability and the thermal and structural properties of shrimp parts | Shell loosening was induced by high pressure heating, while severe denaturation of shrimp meat proteins was caused by ohmic heating. Extreme ohmic heating led to shell tightening, caused by cuticular and epidermal collagen gelatinization. | [72] |
| Ohmic | Tuna fish cubes | Thawing | 40, 50, and 60 V | Study the physico-chemical changes in tuna fish during thawing under ohmic heating | Ohmic heating at 50 V had the shortest thawing time, and significantly decreased thawing loss and total loss compared to traditional thawing, but the samples could be oxidized faster. | [73] |
| Microwave | Surimi gel fortified with fish oil | Heating | (1) 40 °C for 30 min setting + 90 °C for 20 min; (2) 40 °C for 30 min + microwave heating for 96 s (power intensity: 5 W/g) | Investigate the gelation mechanism and physicochemical changes of surimi protein-fish oil composite gels heated by microwave energy | Microwave heating improved the gelling properties, enhanced the chemical forces involved in the gel formation, and changed the secondary structures of surimi protein. | [77] |
| Microwave | Surimi paste | Heating | Thickness = 2.0 cm at 2450 MHz | Establish the relationship between T° distribution and gel properties of surimi with different thickness | Surimi heated by microwave had higher gel properties and water holding capacity compared to that heated by the traditional water bath method. Optimum thickness of surimi for microwave heating was found to be 2 cm. | [78] |

Table A1. *Cont.*

| Thermal Processing | Product | Treatment Purpose | Process Condition | Objective | Main Findings | Reference |
|---|---|---|---|---|---|---|
| Microwave | *MytilusChilensis* and *Dosidicusgigas* | Drying | 90, 160, 360, 600, and 750 W | Study the drying kinetics and moisture diffusivity | Compared to the hot air-drying process, the moisture ratio and drying time decreased rapidly with increasing the microwave power. | [158] |
| Microwave | *Mytilusedulis* | Drying | 90, 180, 360, 600 and 800 W | Microwave drying kinetics study | Microwave power levels slightly affected drying kinetics, rehydration characteristics and energy consumptions. 360 W was the best microwave power level, giving the minimum energy consumption. | [159] |
| Microwave | Grass carp meat | Cooking | 600 W, 70 °C. | Study the physical/ chemical properties Evaluation of saltiness perception impact | Cooking in microwave reduced the degree of protein tertiary structures, decreased cooking loss, maintained the compact of meat structure, and improved the saltiness perception, compared to traditional cooking in water bath. | [160] |
| Microwave | Antarctic krill (*Euphausiasuperba*) and white shrimp (*Penaeusvannamei*) | Thawing and heating | Microwave frequency (300–3000 MHz) in T° range (−20 to 20 °C) | Optimize the technology of the microwave process, and analyze the effects of dielectric properties on the heating rate and temperature distribution | Heating rate was improved by the addition of salt. Salt–sucrose mixture addition changed the dielectric properties significantly and quickened both thawing and heating rates. | [161] |
| Microwave/infrared drying | Rainbow trout (Oncorhynchus mykiss) fillets | Drying | Microwave: 90, 180, 270 and 360 W; infrared drying: 83, 104, and 125 W | Compare the impact of microwave and infrared drying methods on the drying rate, time, and color | Moisture content and color of dried fish samples were significantly affected by time and power levels. Moisture content was higher, drying time was shorter, and color was less influenced using the microwave method. | [162] |
| Microwave /Steam methods | Surimi products | Heating | Microwave power was set at 25, 35, or 45 kW | Compare the changes in the quality and sensory characteristics of surimi heated by two different methods | A thickness of approximately 8 mm was found to be the optimal thickness of surimi to absorb microwaves at 2.45 GHz. Microwave heating can provide energy savings of 11.68% compared to traditional water bath heating. | [163] |

**Table A1.** *Cont.*

| Thermal Processing | Product | Treatment Purpose | Process Condition | Objective | Main Findings | Reference |
|---|---|---|---|---|---|---|
| Radio frequency heating | Vacuum-packed Pacific sauries (*Cololabissaira*) in water | Pasteurization | Radio frequency heating at 9 kW and conventional heating using autoclave, preheated at 80 °C, and then heated up to 120 °C for 15 to 45 min | Inactivate heat-resistant *B. subtilisspores* inside the food without surface cold spots in minimal heating time | *B. subtilisspores* were decreased by 5 logarithmic orders using radio frequency heating. | [164] |
| Radio frequency heating | Vacuum-packed saury in water | Heating | Radio frequency alternating-current energy (9 kW) until the T° reached 120 or 125 °C | Investigate the impact of radio frequency heating on softening and collagen in fish backbone, and characterize the elasticity, crude protein, collagen, and collagen profile of backbone | Water was heated 2.9 times faster than conventional methods. Radio frequency heating achieved wholly edible fish containing low-molecular collagen peptide in a short heating time. | [165] |
| Water bath and ohmic heating | Surimi-canned corn | Cooking | -Water bath at 90 °C for 30 min. -Water bath at 25 °C for 2 h of setting followed by 90 °C for 30 min -Ohmic at 250 V and 10 kHz, for ~30 s at 15 to 90 °C. | Investigate the impact of moisture migration of canned corn mixed with surimi and study texture changes during cooking | Surimi gel texture was influenced by the rate of ohmic heating. Ohmic heating was found to reduce the moisture loss and preserve the texture of surimi gels and corn when compared to the water bath. | [166] |

**Table A2.** Applications of HPP technology on various seafood products.

| Material | Treatment Purpose | Process Condition | Main Findings | Reference |
|---|---|---|---|---|
| Marinated herring | Inactivation of *Morganellapsychrotolerans*, total psychrophilic count, $H_2S$-producing bacteria | 100, 300, and 500 MPa for 5 and 10 min | No microbial growth in samples with 300 MPa for 10 min and 500 MPa for 5, 10 min. Psychrophilic bacteria growth was not detected in samples treated with 500 MPa. $H_2S$-producing bacteria were not observed during 3 months of storage period. | [167] |

**Table A2.** *Cont.*

| Material | Treatment Purpose | Process Condition | Main Findings | Reference |
|---|---|---|---|---|
| Razor clam (*Sinonovaculaconstricta*) | Inactivation of total viable counts | 400 MPa/10 min/46 °C | 5 log reduction | [168] |
| Cold smoked salmon | Inactivation of *Listeria innocua* | 450 and 600 MPa for 120 s at 4 °C | 2.63 and 3.99 $\log_{10}$ CFU/g, respectively | [169] |
| Herring fillets | Inactivation of *Photobacterium phosphoreum* and *Morganellapsychrotolerans* | 100, 200, 300, and 500 MPa for 5 min | 200 MPa did not significantly affect the microbial growth, 500 MPa pressure treatment significantly delayed the growth of *P. phosphoreum* and *M. psychrotoleransuntil* 12th and 7th days of the storage, respectively, 300–500 MPa inhibit the growth of all psychrophilic microorganisms until 19th day of the storage | [170] |
| Salmon, cod, and mackerel | Inactivation of aerobic bacteria count, $H_2S$-producing bacteria | 200 and 500 MPa for 120 s at 8–9 °C | Undetectable level of aerobic counts in cod and mackerel and $H_2S$-producing bacteria in all fish when treated at 500 MPa | [171] |
| Shucked abalone meat | Inactivation of aerobic plate count | 100 and 300 MPa for 5 or 10 min | Extending the shelf-life of 300 MPa treated samples to 35 days as compared with 14 days for controls and for 100 MPa treated samples | [172] |
| Hilsa fillets | Inactivation of total plate count | 250, 350 MPa for 10 min at 27 °C | 2.21 and 2.4 log cycles, respectively; 350 MPa increased the shelf-life to more than 25 days | [173] |
| Smoked rainbow trout and fresh catfish fillets | Inactivation of *Listeria monocytogenes*, *Escherichia coli* | 200, 400 or 600 MPa for 1 or 5 min at room temperature | >6 $\log_{10}$ CFU/g reductions in both fish products | [174] |
| Oyster homogenate | Inactivation of *Vibrio parahaemolyticus* | 200 or 250 MPa for 5 min at 15, 5 and 1.5 °C | Decreasing the population for treatment at 250 MPa for 5 min and 5 °C to 6.2-7.4 $\log_{10}$ CFU/g; same treatment at 1.5 °C showing non-detectable (<10 CFU/g) levels | [175] |
| Chilled mackerel | Inactivation of aerobic mesophilic and psychrophilic count, $H_2S$-producing bacteria | 450 and 550 MPa for 3 and 4 min at 20 °C | Significant extension of microbiological the shelf-life to 29 days and 40 days at 450 MPa/3 min and 550 MPa/4 min, respectively, as compared with 6 days for the control | [176] |

**Table A3.** Applications of ultrasound technology on various seafood products.

| Material | Treatment Purpose | Process Condition | Main Findings | Reference |
|---|---|---|---|---|
| *Labeorohita* head | Ultrasound- assisted extraction (UAE) of oil | UAE: 20 kHz, 40% amplitude, for 5, 10 and 15 min. Enzymatic hydrolysis: Protamex ratio of 1:100 (*w/w*), 2 h, 150 rpm, 55 °C | Pretreatments with UAE improved the extraction yield of oil, showing higher oil recoveries (67.48% vs. 58.74% for SFE and untreated samples, respectively) | [177] |
| Bighead carp (*Hypophthalmichthysnobilis*) scales | Ultrasound- assisted extraction of gelatin | Temperature: 60, 70 and 80 °C Extraction time: 1 h | Improved technological properties: highest storage modulus (5000 Pa), gelation point (22.94 °C), and melting point (29.54 °C) | [178] |
| Bighead carp (*Hypophthalmichthysnobilis*) scales | Ultrasound- assisted extraction of gelatin | Temperature: 60 °C Extraction time: 1, 3 and 5 h | Extraction yield: 46.67% for ultrasound bath versus 36.39% for water bath | [179] |
| Mackerel | Ultrasound- assisted extraction of proteins | ISP: Isoelectric solubilization precipitation. UAE: 40 kHz, 60% amplitude, 0.1 M NaOH, 10 min. | Significant increase of protein recovery, recovering more than 95% of total protein from mackerel by-products | [180] |
| Raw salmon | Inhibition of *Listeria monocytogenes* | Ultrasound of bath, 45 kHz, 200 W, 1 min | 0.35/g Decimal reduction (log cfu) and antimicrobial effects | [181] |
| Red seabream (*Pagrus major*) fillets | | Ultrasound and thawing at 0 °C under vacuum (UVT) (40 kHz, 200 W, 10 °C) | No free water changes and improved physicochemical properties of proteins, actin | [182] |
| Cod (*Gadus morhua*) fillets | | Ultrasound and hydration medium's pH (from 8.5 to 10.5) (1) 25 kHz, 29.4 W/kg, 113,7 W, 20 min, 14 °C (2) 25 kHz, 14.7 W/kg, 64.3 W, 20 min, 14 °C (3) 25 kHz, 2.9 W/kg, 15.3 W, 20 min, 14 °C | 2.9 W/kg: produced the highest increments in WG (18.6%), reducing hydration time by 33% US+pH 8.5: 1-day shorter hydration time, US+pH 8.5: improved microbial quality | [183] |
| Brown crab (*Cancer pagurus*) whole cooked | | Ultrasound and freeze drying at −20 °C, 20 kHz, 400 W, 0, 5, 10, 15, 20 min, room temperature | Allergenicity decreased with increasing treatment time (tropomyosin reduced 76% after 20 min of US treatment) Total antioxidant capacity strengthened | [184] |

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
