# Peer review of "Seafood Processing, Preservation, and Analytical Techniques in the Age of Industry 4.0"

_applsci, doi:10.3390/app12031703_

Round 1

Reviewer 1 Report

The authors did a good work from an experimental point of view, and I recommend the article for publication. Only I have some format corrections.

More specific in Table 3:

Fishbone… correct ‘’9 pulse,’’ by ‘’9 pulses’’.

Haliotis discus… ‘’(20 kV/cm’’ delete the parenthesis.

Residues of rainbow… correct ‘’1–3 kV.cm-1’’ by ‘’1–3 kV/cm’’.

Author Response

Comments and Suggestions for Authors

The authors did a good work from an experimental point of view, and I recommend the article for publication. Only I have some format corrections.

More specific in Table 3:

Fishbone… correct ‘’9 pulse,’’ by ‘’9 pulses’’.

Haliotis discus… ‘’(20 kV/cm’’ delete the parenthesis.

Residues of rainbow… correct ‘’1–3 kV.cm-1’’ by ‘’1–3 kV/cm’’.

Our response: We thank the reviewer for taking the time to review our manuscript. We are glad to hear that the reviewer finds our work good.

Fishbone… correct ‘’9 pulse,’’ by ‘’9 pulses’’.

Our response: Thank you for pointing out this error. The correction has been made.

Haliotis discus… ‘’(20 kV/cm’’ delete the parenthesis.

Our response: Thank you for referring to this mistake. This has been fixed.

Residues of rainbow… correct ‘’1–3 kV.cm-1’’ by ‘’1–3 kV/cm’’.

Our response: Again thank you for this correction.

Reviewer 2 Report

In this paper authors gave an overwiev of the impact of 4IR on the seafood processing and preservation. Althoug the article could be intresting to the readers, the paper is to long and there is a lot of unnecessary repeating of the same informations. The description of each IR phase is unnecessarily  long, especially because their carachteristics are sumarized in Table1. Further more, description of current and emerging trends is too long and it must be shortened . Figures 1 and 3 can be excluded. 

Minor:

  1. In lines 68 and 69 it is claimed that the quality of the food is better due to 4th ind. revolution. Can you specify the parameters against which food quality was assesed?
  2. Line 112 - name of the first author should be stated before the number of citation.

Author Response

In this paper authors gave an overwiev of the impact of 4IR on the seafood processing and preservation. Althoug the article could be intresting to the readers, the paper is to long and there is a lot of unnecessary repeating of the same informations. The description of each IR phase is unnecessarily  long, especially because their carachteristics are sumarized in Table1. Further more, description of current and emerging trends is too long and it must be shortened . Figures 1 and 3 can be excluded. 

Our response: We thank the reviewer for taking the time to review our manuscript. All the reviewer’s suggestions and recommendations have been taken into consideration. The description of IR as well as the sections about current and emerging trends been shortened. However we think that the figures are useful as it is important to present information to the reader in a visually appealing way. Figure 1 gives a general overview of applications of smart sensors, based on hyperspectral imaging in the seafood sector, while Figure 3 sums up relevant aspects related to the mass spectrometry.  

Minor:

  1. In lines 68 and 69 it is claimed that the quality of the food is better due to 4th ind. revolution. Can you specify the parameters against which food quality was assesed?

Our response: We thank the reviewer for this question. The main advantage of implementing 4IR elements is reducing production time and cost and providing better quality. One example can be the use of smart sensors for monitoring and sorting products during processing and manufacturing processes. Such sensors can be directly placed on production lines to assess quality and sort seafood or other food products into different quality grades. A short sentence has been added to the text to explain this case.

2. Line 112 - name of the first author should be stated before the number of citation.

Our response: Thank you for this comment. The citation has been corrected in the revised version. 

Round 2

Reviewer 2 Report

Authors made all the required corrections and paper is acceptable for publication now.